# Dynamic Coupling of Tyrosine 185 with the Bacteriorhodopsin Photocycle, as Revealed by Chemical Shift Assisted AF-QM/MM Calculations and Molecular Dynamic Simulations

**DOI:** 10.3390/ijms222413587

**Published:** 2021-12-18

**Authors:** Sijin Chen, Xiaoyan Ding, Chao Sun, Anthony Watts, Xiao He, Xin Zhao

**Affiliations:** 1School of Physics and Electronic Science, East China Normal University, 500 Dongchuan Road, Minhang District, Shanghai 200241, China; 52194700015@stu.ecnu.edu.cn (S.C.); xiaoyan.ding@bioch.ox.ac.uk (X.D.); csun@phy.ecnu.edu.cn (C.S.); 2Department of Biochemistry, University of Oxford, South Park Road, Oxford OX1 3QU, UK; anthony.watts@bioch.ox.ac.uk; 3Shanghai Engineering Research Center of Molecular Therapeutics and New Drug Development, School of Chemistry and Molecular Engineering, East China Normal University, Shanghai 200062, China; 4NYU-ECNU Center for Computational Chemistry at NYU Shanghai, Shanghai 200062, China

**Keywords:** bacteriorhodopsin, Tyrosine 185, retinal chromophore, photo-intermediate and photocycle, AF-QM/MM calculations and MD simulations

## Abstract

Aromatic residues are highly conserved in microbial photoreceptors and play crucial roles in the dynamic regulation of receptor functions. However, little is known about the dynamic mechanism of the functional role of those highly conserved aromatic residues during the receptor photocycle. Tyrosine 185 (Y185) is one of the highly conserved aromatic residues within the retinal binding pocket of bacteriorhodopsin (bR). In this study, we explored the molecular mechanism of its dynamic coupling with the bR photocycle by automated fragmentation quantum mechanics/molecular mechanics (AF-QM/MM) calculations and molecular dynamic (MD) simulations based on chemical shifts obtained by 2D solid-state NMR correlation experiments. We observed that Y185 plays a significant role in regulating the retinal *cis–trans* thermal equilibrium, stabilizing the pentagonal H-bond network, participating in the orientation switch of Schiff Base (SB) nitrogen, and opening the F42 gate by interacting with the retinal and several key residues along the proton translocation channel. Our findings provide a detailed molecular mechanism of the dynamic couplings of Y185 and the bR photocycle from a structural perspective. The method used in this paper may be applied to the study of other microbial photoreceptors.

## 1. Introduction

Bacteriorhodopsin (bR), a microbial rhodopsin photoreceptor found in the purple membrane of *Halobacterium salinarum*, is a desirable system for studying the mechanism of energy transduction and ion transport in biological membranes. It consists of seven-transmembrane (7TM) helices and acts as a light-driven proton pump transporting protons against pH gradient across the membrane from the cytoplasmic side to the extracellular side [1,2,3]. The retinal chromophore is covalently bound to a lysine residue (K216) on helix G to form a protonated Schiff base (SB) linkage and possesses a thermal equilibrium between 13-*cis*, 15-*syn* (denoted as bR*_cis_*), and all-*trans*, 15-*anti* (bR*_trans_*) isomers in the dark-adapted bR. Photoisomerization of the all-*trans* retinal to the 13-*cis* isomer induces a series of changes in the receptor to realize the transferal of a proton across the cell membrane through a photocycle including K, L, M_1_, M_2_, M_2_’, N, and O intermediate states [4,5,6,7,8,9,10,11,12,13,14,15,16]. A detailed description of the bR photocycle and proton transfer steps is shown in Figure 1.

Tyrosine 185 (Y185), one of the most conserved aromatic residues among the microbial rhodopsin family proteins, is located within the retinal binding pocket in helix F of bR (Appendix A). It is reported that Y185 regulates the bR photocycle, such as stabilizing the hydrogen-bond (H-bond) network on the extracellular side of SB and involving the deprotonation of SB during the early stage of the photocycle [17,18]. Neutron scattering experiments have presented an excellent dynamic picture of a more rigid retinal binding pocket and an extracellular half of the membranes compared to the membrane globally, as well as the existence of two dynamic transitions in the fully hydrated bR [19,20,21,22,23]. Our recent studies have revealed that Y185 stabilizes the retinal *cis–trans* isomerization thermal equilibrium in the dark-adapted state by an H-bond interaction with the SB N–H group. Removal of the phenolic hydroxyl in Y185F shifts retinal *cis–trans* thermal stability to the bR*_cis_*-dominated state and affects the bR photocycle kinetics and the ATP formation rate [24,25]. However, a detailed description of the dynamic coupling of Y185 with the bR photocycle in the different intermediate states at a molecular level is still necessary.

Molecular dynamic (MD) simulations have provided a powerful tool to probe the dynamic conformation of membrane proteins ranging from nanometers close to micrometers on a picosecond to a millisecond timescale [26,27,28]. In the 1980s, the first simulations of a pure lipid bilayer were carried out; shortly afterwards, the bilayer membrane protein was simulated by continuous medium electrostatics [29]. By the 1990s, Woolf et al. had applied an explicit phospholipid model to the MD simulations of membrane proteins for the first time [30]. At the same time, several hundred picosecond simulations of the light-driven proton pump bR were carried out under vacuum conditions [31]. Currently, MD simulations of membrane proteins can reach the millisecond timescale [32]. The specific role of crucial residues, conformational dynamics, and activation mechanisms of membrane proteins can be investigated at the molecular level in detail [24,33,34,35]. 

Generally, the initial structure constructed by the homology model for the MD simulations of a protein has a significant impact on its results, and accurate construction of the ligand and its binding pocket can improve the quality of the simulated protein structure. In 2000, Cui and Karplus developed a quantum mechanics/molecular mechanics (QM/MM) approach to calculate the NMR chemical shifts molecularly [36]. Subsequently, Gao and colleagues proposed the adjustable density matrix assembly (ADMA) [37], and Frank and colleagues presented the fragment molecular orbital (FMO) approach to improve chemical shift calculations by taking into account various conditions [38,39]. Later, he and his colleagues proposed a more efficient automated fragmentation QM/MM approach (AF-QM/MM) for routine ab initio NMR chemical shift calculations for proteins of any size [40,41,42,43]. 

In this paper, we use a refined retinal binding pocket as the starting point for the MD simulation of bR, based on the AF-QM/MM calculations of the chemical shifts obtained by the solid-state NMR (ssNMR) correlation experiments of the ^13^C-retinal-incorporated purple membrane. We then explore the molecular mechanism of dynamic coupling of Y185 with some of the key events during the bR photocycle in wild-type (WT) bR and its Y185F mutant (phenylalanine replaced tyrosine). Our results show that Y185 plays a significant role in regulating the retinal *cis–trans* thermal equilibrium, stabilizing the pentagonal H-bond network, participating in SB nitrogen flipping, and opening the F42 gate. This mechanistic study reveals the detailed molecular mechanism of Y185’s function in the bR photocycle from a structural point of view. In addition, the method adopted in this work could apply to the studies of the roles of the key residues in other microbial photoreceptors. Appendix A summarizes the flowchart of the calculations, and Appendix A summarizes the 14 simulation models used.

## 2. Results and Discussion

### 2.1. Coupling of Y185 with the Retinal Chromophore in the Dark-Adapted bR

Table 1 shows the dark-adapted and M state chemical shifts of the [10, 11, 14, 15-^13^C_4_]-retinal-regenerated wild-type bR (WT-bR) and its Y185F mutant (Y185F-bR), as determined by ssNMR [25] and calculated by the AF-QM/MM method in this work. The root–mean–square error (RMSE) of 0.5 between the experiments and the AF-QM/MM calculations indicates a remarkable convergence and matches the experimental results well. This calculated structure of the retinal chromophore will provide a more precise retinal binding pocket for the MD simulation of the overall system at a later stage.

The all-atoms MD simulations of the membrane-embedded WT- and Y185F-bR were carried out in the 1-palmitoyl-2-oleoyl-sn-glycerol-3-phosphocholine (POPC) lipid bilayer for 100 ns with the implanted geometry of the retinal binding pocket calculated by AF-QM/MM as the initial structure for each system. The results were verified by the root–mean–square deviation (RMSD) of the backbone Cα atom of the protein relative to the starting coordinates of the crystal structure. As seen in Figure 2A and Appendix A, the RMSDs of all simulated models converged after 20 ns and remained below 2.5 Å. Positions and orientations of the retinal binding residues in WT-bR*_trans_* and WT-bR*_cis_* are mostly similar to those in the corresponding crystal structures. For example, our simulation shows the L93 side chain to C20 of the retinal is 3.86 ± 0.29 Å in WT-bR*_trans_*, 3.8 Å in the crystal structure of 1C3W [44] and 3.9 Å in 1BRR [45]. W182 to C20 of the retinal is 4.13 ± 0.35 Å in our simulation of WT-bR*_cis_*, and 4.0 Å in the crystal structure of 1X0S [46]. Appendix A summarize the other distances between the retinal and the binding residues or those among themselves. RMSD of the wild-type protein was slightly larger than Y185F in the dark-adapted form because the loop region of the wild-type may fluctuate more (Figure 2A and Appendix A). In short, the stable trajectories and positions of the residues with little difference between simulated and crystal structures indicate that the hybrid templates used here were reasonable and could be employed further to analyze the dynamic function of Y185 during the photocycle.

Compared with the crystal structures of bR*_trans_* and bR*_cis_*, it is worth mentioning that the distance between the phenolic hydroxyl of Y185 and the SB nitrogen varied significantly: 3.3 Å in the simulated WT-bR*_trans_* but 4.8 Å in the crystal structures of 1C3W and 1BRR; 3.2 Å in the simulated WT-bR*_cis_* but 4.9 Å in the crystal structures of 1X0S (Figure 2D). Such a short distance may imply the formation of an H-bond interaction between Y185 and SB in the dark-adapted protein (Figure 2E), which is consistent with the downfield shifted chemical shift of the Cζ atom of Y185 in our previous paper [24]. Compared with the dark-adapted WT-bR, the retinal and K216 moved considerably in the Y185F. Our MD simulations showed that the SB nitrogen moved 1.9 Å in the cytoplasmic direction with an even twisted C12–C13 bond and flattened C14–C15 and C15=Z bonds in Y185F-bR*_trans_*. However, the different configuration observed in Y185F-bR*_cis_* was more flattened from the β-ionone ring to the C12–C13 bond, and more twisted in the SB region (C14–C15 and C15=N bonds), as depicted in Figure 2B and Appendix A. These simulation results again suggest the existence of different interaction modes of Y185 and F185 with the retinal chromophore. All simulations were repeated twice on WT-bR*_trans_*, Y185F-bR*_trans_*, WT-bR*_cis_*, and Y185F-bR*_cis_* to rule out any statistical incident from only one simulation. Similar H-bond interactions between Y185 and SB in WT-bR*_trans_* and WT-bR*_cis_*, as well as changes in the retinal sidechain torsion angles, were observed from both simulations, as shown in Appendix A. 

The influence of this H-bond on the retinal binding pocket was then further investigated by Y185F mutation. Figure 3A–D display the simulation results of the retinal binding pocket of WT- and Y185F-bR in the dark-adapted forms. For Y185F-bR*_trans_*, the benzene ring of F185 is essentially unchanged; W86, W189, D85, and M145 lean toward the polyene chain of the retinal; W182 and L93 move away from the retinal with a small displacement (Figure 3A, Appendix A). For Y185F-bR*_cis_*, F185, W189, and W86 moved away from the retinal, while M145, W182, and L93 slightly titled towards the retinal (Figure 3B, Appendix A). Furthermore, the cytoplasmic sides M145 and L93, and the extracellular sides W189, Y83, W86, and D85 in the retinal binding pocket of Y185F-bR*_trans_* overlapped with those corresponding residues in WT-bR*_cis_*, but the rearrangements of these residues in Y185F-bR*_cis_* were different (Figure 3C,D). As the retinal binding pocket volume size of bR reflects its preference to the retinal configuration to some extent [47,48,49], the polyhedral volumetric model of the pocket detection plugin (Pck 2.0.3) of VMD [50,51] was used to calculate the binding pocket volumes of WT-bR*_trans_*, WT-bR*_cis_*, Y185F-bR*_trans_*, and Y185F-bR*_cis_* in the dark-adapted forms. Different pocket volumes of 947.67 Å^3^, 968.09 Å^3^, 883.40 Å^3^, and 989.80 Å^3^ were obtained for the retinal in either *cis-* or *trans*-configurated WT-bR and Y185F-bR, respectively (Table 2 and Appendix A). The pocket volume of WT-bR*_cis_* was smaller than that of WT-bR*_trans_*, and the binding pocket volume of Y185F-bR*_trans_* distinctly reduced close to that of WT-bR*_cis_* with no significant change in Y185F-bR*_cis_* (Table 2 and Appendix A), which are consistent with the argument that a decrease in the retinal binding pocket volume is favorable for the 13-*cis* isomer [47,48,49]. These observations indicate that the retinal binding pocket of Y185F-bR is more suitable for the *cis*-configurated retinal chromophore and clearly explain why the ssNMR measurements showed a Y185F-bR*_cis_* dominated form in the dark-adapted mutant, as reported previously [24]. Mediation of the H-bond to the retinal binding pocket further affected the conformation dynamics of the whole protein, as illustrated by the dynamic cross-correlation map (DCCM). WT-bR*_trans_* and WT-bR*_cis_* showed a more positive correlation, indicating a more coherent protein dynamic conformation. Y185F-bR*_trans_* and Y185F-bR*_cis_* presented a more negative correlation, implying a more incoherent protein dynamic conformation (Figure 3E–H). Clearly, different couplings of the retinal *cis–trans* thermal equilibrium with the overall protein dynamics exist in the wild-type and mutated proteins. Removal of the Y185–O·····SB–NH H-bond disturbed the dynamic feature of the whole protein and eventually weakened its proton translocation function.

### 2.2. Regulation of the Extracellular Side Pentagonal H-Bond Network of the Retinal Binding Pocket 

The pentagonal H-bond network is composed of SB, D85, D212, R82, and three water molecules (Wat401, Wat402, Wat406) in the extracellular side of the retinal binding pocket, and it is crucial for the proton transfer in bR [44,52] (Figure 4A). Our initial study reported that Y185 might regulate this pentagonal H-bond network and influence the proton pumping function [24]. To further explore the molecular mechanism of how Y185 regulates the pentagonal H-bond network, MD simulations of WT-bR and its Y185F mutant were performed based on the initial crystal structure of 1C3W [44].

Our simulations showed that removal of the phenolic hydroxyl group in Y185F causes T89 and D85 to move away from each other. The mean distance between the hydroxyl group of T89 and the carboxyl group of D85 is 2.74 ± 0.16 Å in the wild-type protein and 3.82 ± 0.51 Å in Y185F in the last 10 ns simulations, as shown in Figure 4D. Since the pKa of D85 depends partially on the H-bond strength between T89 and D85 [7,53,54], increasing this distance between T89 and D85 will weaken the H-bond formation and increase the pKa of D85 in Y185F, as we observed previously [25]. In addition, the number of water molecules within 4.0 Å around D85, D212, R82, and SB remained 3 to 4 in WT-bR (Figure 4B), but 2 and 3 in Y185F-bR in our MD simulations, respectively (Figure 4F). The number of frames with three water molecules accounted for about 70% of the total simulated structures, and four water molecules accounted for the remaining 25% (Figure 4C). On the other hand, two and three water molecules each accounted for 40% of the total number of trajectories in Y185F-bR (Figure 4G). These observations indicate that the Y185F mutation may disrupt the pentagonal H-bond network and form a new H-bond network with fewer H-bonds between Wat402 and SB, Wat402 and D212, Wat402 and D85, and Wat401 and D212 (Figure 4E,H). Our second-time simulation also showed similar results (Appendix A). This speculation agrees well with the previous measurements suggesting that the Y185 mutant causes a weakening absorption strength of the O-D involved in the pentagonal cluster [55].

### 2.3. Coupling of Y185 with the M_1_ and M_2_ States

Generally, the M state can be divided into M_1_ (early M), M_2_ (late M), and M_2_’ (terminal M), three substates in the bR photocycle to differentiate its fine structure and protonation condition of several key residues [4,5,6,16,56,57,58]. The protonated SB becomes deprotonated when arriving at the M_1_ substate. The lone-pair-electron of the SB nitrogen points to the extracellular side till to the M_2_ substate and releases a proton to the extracellular surface in the M_2_ to M_2_’ transition. To gain deeper insight into the role of Y185 in the M state, we performed MD simulations on the M_1_ and M_2_ substates of WT-bR and Y185F based on the chemical shifts obtained by the ssNMR experiments [25].

Our simulations showed that the M_1_ substate could be further divided into two phases, as indicated by Figure 5A. The first phase is that the lone-pair-electron of the SB nitrogen points to the extracellular side (denoted as M_1_-es), and the second phase is that the lone-pair-electron of the SB nitrogen flips to point to the cytoplasmic side (M_1_-cs). The residence time was 20 ns for the WT-M_1_-es and 8 ns for the Y185F-M_1_-es, indicating that SB nitrogen was more prone to reorienting in the Y185F mutant. The retinal binding pocket volume of the Y185F-M_1_-es (985.84 Å^3^) was more significant than that of the WT-M_1_-es (944.61 Å^3^) (Table 2) to ensure the fast reorientation of the SB nitrogen. Compared with the WT-M_1_-es, the binding pocket and channel residues in the Y185F-M_1_-es had apparent displacement (Figure 5B,C, Appendix A). F185, W86, and Y57 moved away from the retinal chromophore to a certain degree on the extracellular side. Significant upward movement of the retinal sidechain from C9 to NZ and the SB nitrogen occurred towards the cytoplasmic direction. The C13 methyl group of the retinal moved upwards, resulting in an evident shift in W182, L93, and M145. L93 moved away from the C13 methyl group, whereas the distance between the W182 and C13 methyl groups remained unchanged. Overall, the binding pocket volume of the Y185F-M_1_-es increased with large rearrangements in the β-ionone binding region, which indicates that Y185 may support forming the M_1_ substrate and reorienting the SB nitrogen (Table 2). In the Y185F-M_1_-cs, the β-ionone showed a notable change with the displacements of W138 associated with W189 (Figure 5D, Appendix A); however, the rest of the binding residues were virtually unchanged. The binding pocket volume was 977.91 Å^3^ for WT-M_1_-cs, and 985.53 Å^3^ for Y185F-M_1_-cs, respectively (Table 2), indicating that Y185F mutation does not significantly affect the binding pocket of the M_1_-cs. In the M_2_ state, Y185F did not cause much of a change in the conformation of the proton release complex (PRC), despite the distance of D212 to the benzene ring of F185 increasing. Y57 and D212 remained intact, and the spaces of Y83 to E194 and E204 to E194 changed slightly; only the guanidine group of R82 pointed more towards E204 due to the carboxyl of E204 rotating (Figure 5E,F, Appendix A). These observations imply that no noticeable changes occurred to the H-bond network around the proton release region (Appendix A). The conformation change caused by the Y185F mutation had almost no impact on the proton release process, which is in line with our previous experimental results [25]. 

### 2.4. Participation in Opening the F42 Gate in the M_2_’ to N States

Conformation dynamics are the fundamental feature of any membrane protein in general, allowing for a rapid response between distant structural domains by various interactions to fulfil its dynamic function. For example, Bondar et al. proposed that perturbations caused by mutations in remote regions of the SecY Translocons could be relayed to the plug rapidly, causing its displacement and increasing hydration [59]. Tanio et al. demonstrated that the D85-V49 long-distance effect impacted bR backbone conformation [60]. Therefore, MD simulations were further carried out on WT-M_2_’, WT-bR_N_, Y185F-M_2_’, and Y185F-bR_N_ to explore the effect of Y185 on the M_2_’-N transition. All simulations were based on the bR crystal structures of M_2_’ (1C8S) and N (4FPD) states [7,10]. 

Our simulations showed that F42 and Y43, two residues near the cytoplasmic surface, changed significantly in Y185F-M_2_’. In general, the benzene ring of Y43 displaced towards helix C, the benzene ring of F42 turned at least 90° inwards to partially cover the D96 and block the access of the continuous water chain from the cytoplasmic surface to SB (Figure 6A,B and Appendix A). Furthermore, we found that the orientation of F42 in Y185F-bR_N_ is the same as that in the WT-M_2_’ and WT-bR_N_, and the continuous water chain exists from the cytoplasmic surface to SB (Figure 6C,D and Appendix A). As we know that F42 acts as a dynamic gate to control water into the protein from the cytoplasmic surface to SB [61,62], our observations suggested that Y185 may have a long-range influence on the flipping of the F42 benzene ring to ensure its function.

The molecular mechanism of how Y185 regulates F42 remotely could be monitored through the motions and interactions (H-bond and hydrophobic interactions) of those key residues along the cytoplasmic half-channel from F185 to F42 in Y185F-M_2_’ (Figure 6G). In the Y185F-M_2_’, the retinal C13-methyl group moved upward due to the disappeared H-bond interaction of F185 with D212 (Appendix A, and step 1 in Figure 6G). Then, this upward movement pushed W182 to move up (Appendix A, and step 2 in Figure 6G), which further caused a sidechain rotation of L93 and a movement of F219 (Appendix A). In the Y185F-M_2_’, D96 formed a water-mediated H-bond with T46 instead of a direct H-bond in the WT-M_2_’ (Figure 7A,B, Appendix A, and step 3 in Figure 6G). Those conformation changes finally caused the cytoplasmic end of helix B to shift towards helix C by at least 1.1 Å and broke the backbone H-bonds of F42 with D38 and Y43 with A39 (Figure 6E,F, step 4 in Figure 6G). As a result, Y43 had H-bond interactions with three water molecules in Y185F-M_2_’ instead of the hydrophobic interactions with G31 and F27 in WT-M_2_’ (Figure 7C,D), which caused it to move towards F42 and pushed it to leave its original position as in the WT-M_2_’ (Figure 6G). The displacement of helix B caused by the new hydrophobic interactions of F42 with L99 and L100, along with the disappearance of the hydrophobic interaction with G220 (Figure 7E,F), was the main driven force to keep F42 in the new position in the Y185F-M_2_’ (step 5 in Figure 6G). Our findings here are in good agreement with the previous studies showing that shifting W182 disrupted various interactions in the cytoplasmic side [7,8,63,64]. In a word, coupling of Y185 with D212 initiates a series of conformation rearrangements along the cytoplasmic half-channel to remote control the F42 gating in WT-M_2_’. Similar evolution patterns of those key distances were observed throughout the whole simulations of WT-M_2_’ and Y185F-M_2_’, as shown in Appendix A. The standard deviation differences of those crucial distances from the retinal binding pocket to the cytoplasmic half-channel by the last 10 ns simulation and throughout the whole simulation may imply a good preference for the conformation change starting from Y185 to F42 (Appendix A), as we proposed in Figure 6G.

In the Y185F-M_2_’, F42 flipped to block the continuous passage of water from the cytoplasmic surface to SB. However, in the Y185F-bR_N_, the F42 gate eventually opens to reestablish the constant water chain and reprotonation of D96 and SB. These observations suggest that Y185F slows down the opening process of F42 and prolongs the construction of the water chain in the cytoplasmic half channel. Although Y185 and F42 are far from each other, the perturbation of the extracellular side to the cytoplasmic side is unfavorable. Luecke et al. proposed that M decay occurs when several residues and water molecules in the cytoplasmic side form a transient proton-conducting network from the surface to SB [7]. del Val et al. reported that the retinal binding residue D212 has long-range effects on water and hydrogen-bonding dynamics of the cytoplasmic side, and the D96-T46T47 (DTT) motif is more sensitive to long-range perturbations [35]. Ding et al. reported that the Y185F mutation prolonged the M decay [25]. Therefore, the prolonged M state decay mechanism in Y185F may be attributed to the long-range coupling of Y185 with F42 via the DTT motif in the second half of the photocycle. To rule out any statistical incident from only one simulation, we repeated the simulations for WT-M_n_ and Y185F-M_n_ three times. All repetitive simulations showed that the F42 gate opened in WT-M_n_ and closed in Y185F-M_n_. Figure 8A,B compared the cytoplasmic half channel of the WT-M_2_’ and Y185F-M_2_’ from the second-and the third-time simulations. Figure 8C,D indicated that the repeatability of the two simulations is excellent for both WT-M_2_’ and Y185F-M_2_’.

## 3. Conclusions

This study systematically investigated the functional role of Y185 at the atomic level. We simulated 14 systems of the wild-type and Y185F mutant of bR at different intermediates to monitor the functional role of Y185 with the bR photocycle at the atomic level. We demonstrated that Y185 performs essential functions during the bR photocycle, including regulation of the dark-adapted retinal *cis–trans* thermal equilibrium, stabilization of the pentagonal H-bond network, participation of the orientation switch of the SB nitrogen, and the opening of the F42 gate by interacting with the retinal and several key residues along the proton translocation channel. Our findings are the first to provide a detailed molecular mechanism of the dynamic couplings of Y185 with the bR photocycle from a structural perspective (Figure 9). In addition, the method adopted in this work, which is based on the AF-QM/MM calculated retinal binding pocket for overall MD simulations, can be applied to the studies of other key residues of microbial rhodopsin family proteins. The molecular mechanistic insights into the functional role of the key residue provided here will lead to new opportunities and versatility for proton pump-based optogenetic applications.

## 4. Materials and Methods

### 4.1. Molecular Models and Parameters

The following bR crystal structures were used as templates in our calculations and simulations: the crystal structure of the dark-adapted state with the all-*trans* retinal chromophore (PDB code 1BRR, without internal water) [45]; the crystal structure of light-adapted with the all-*trans* retinal chromophore (PDB code 1C3W, with internal water) [44]; the dark-adapted crystal structure with 13-*cis* retinal chromophore (PDB code 1X0S) [46]; the M_1_ intermediate state crystal structure (PDB code 1M0M) [9]; the M_2_ intermediate state crystal structure (PDB code 1F4Z) [8]; the M_2_’ intermediate state crystal structure (PDB code 1C8S) [7]; the N intermediate state crystal structure (PDB code 4FPD) [10]. The 13-*cis*, M_1_, M_2_ monomer contains 227 residues, the all-*trans* monomer contains 230 residues, and the N monomer contains 225 residues. The partial residues missed in loop regions were filled by using Schrödinger Suites [65]. For the Y185F mutant (Y185F-bR), we mutated tyrosine 185 to phenylalanine by PyMOL [66].

A monomer taken from the X-ray structure of each protein was embedded into a 1-palmitoyl-2-oleoyl-sn-glycerol-3-phosphocholine (POPC) lipid bilayer, the topologies and parameters of which had been extensively tested [67]. The CHARMM-GUI server [68,69,70,71] was used to build the membrane with water. The bR molecule was then placed into POPC, composed of 128 lipids (64 in the upper leaflet and 64 in the lower brochure). Then, 0.15 M sodium chloride, along with 4735 water molecules, was added to neutralize the system, which generated a protein–lipid–water complex (Appendix A). The position of bR in the lipid membrane was determined by the PPM web server [72]. Asp96, Asp115, Glu204, and the Schiff base in the dark-adapted and light-adapted state and Asp85, Asp96, and Asp115 in the M and N states were protonated. 

The final simulated systems were constructed using the XLEAP program [73]. The retinal was treated as a single unit, and the parameters were obtained using the antechamber module and the general force field (GAFF) [74] in the AMBER program. POPC lipid molecules were assigned Lipid14 force filed [75], and the parameters were assigned based on the AMBER ff14SB [76]. The TIP3P model [77] was utilized to construct water molecules. All the MD simulations were carried out using the pmemd.cuda module of the AMBER16 program [78], and 100 ns MD simulation with periodic boundary conditions was conducted in the NPT ensemble at 303 K and 1 bar. Long-range electrostatic interactions were treated using the PME method [79], and empirical Lennard-Jones potentials calculated short-range van der Waals interactions with a cutoff of 10 Å. The temperature was regulated by Langevin dynamics [80] with a collision frequency of 1.0 ps^−1^, the anisotropic Berendsen weak-coupling method [81] was utilized to couple the system to a barostat of 1 bar, and the SHAKE algorithm [82] was used to deal with the vibrations involving hydrogen atoms.

### 4.2. Automatic Fragmentation Quantum Mechanic/Molecular Mechanic Calculations

The chemical shifts in the retinal chromophore at the 10, 11, 14, and 15 sites were calculated by automatic fragmentation quantum mechanics/molecular mechanics method (AF-QM/MM) [40,83]. For the accuracy and efficiency of this method, please refer to the publications by He et al. [40,41,42,43]. Briefly, we divided the whole protein into three regions: core, buffer, and remaining, as shown in Appendix A. The protonated Schiff base, including K216, was defined as the core region. The buffer region was defined by the residues within the retinal binding pocket. Some other residues were found within 2.5 Å of the core region, where the contacting atoms are hydrogen, as well as residues within 4.0 Å, where at least one of the contacting atoms is non-hydrogen. The remaining part of the protein was defined as the rest region. The QM calculations treated the core and buffer regions, and the remaining area was calculated at the MM level (as background charges). The chemical shifts were calculated by Gaussian 09 [84] using the GIAO [85] method at the B3LYP/6-31G** level. To obtain a more accurate retinal binding pocket, we slightly adjusted the retinal bond length, torsion angle, and the binding residues to ensure that the theoretically calculated value and the experimental chemical shift value were within the error tolerance range (RMSE ≤ 0.5).

### 4.3. Molecular Dynamic Simulation

The retinal binding pocket geometry calculated by AF-QM/MM was implanted into the whole protein for the MD simulations. The remaining structure directly used the crystal structure, as it is used for the all-atom simulations. Molecular dynamic simulations were carried out using the GPU-CUDA version AMBER16 program package [78]. Firstly, we performed the simulations with everything fixed, except lipid tails, to induce the appropriate disorder of a fluid-like bilayer. Then, each of the systems needed two-stage energy minimization. In the first stage, the protein, crystal waters, and lipids were restrained by a harmonic potential with a force constant of 10 kcal/mol·Å^2^ while all other atoms were unrestrained; in the second stage, all atoms were relaxed. Energy minimization was executed for 100,000 steps per stage. The steepest descent method was used for the first 50,000 steps, which was then switched to the conjugate gradient method for the rest of the steps. After energy minimization, the whole system carried out three stages of equilibration. In the first stage, the systems were gradually heated from 0 to 100 K at 500 ps in the NVT ensemble, in which the bR protein and crystal waters were restrained by a harmonic potential with force constant of 5.0 kcal/mol·Å^2^, the lipids were controlled by a force constant of 2.5 kcal/mol·Å^2^; in the next stage, the systems were gradually heated from 100 to 300 K at 500 ps in the NPT ensemble which the constant pressure was 1.0 bar; in the third stage, for 1 ns, the restraints of force constant on the bR protein and crystal waters reduced to 1.0 kcal/mol·Å^2^, and the force constant on lipids reduced to 0.5 kcal/mol·Å^2^. To maintain the refined configuration of the retinal, the atoms calculated by the QM method were fixed with a force constant of 999.0 kcal/mol·Å^2^. Finally, the direct MD simulations were carried out with all the atoms free, and the coordinates were saved every 4 ps. 

### 4.4. Analysis of the Simulations

The cpptraj module of AmberTools18 was used to calculate the root–mean–square deviations (RMSD), root–mean–square fluctuation (RMSF), dynamic cross-correlation map (DCCM), the distances between key residues, the dihedral angle of the retinal, and the hydrogen-bond. The hydrogen-bond and the van der Waals and hydrophobic interactions in the M_2_’ state were analyzed using HBPLUS [86] and LIGPLOT [87]. The pocket volume changes in bR in different intermediate states were calculated by the pocket detection plugin of VMD [50,51]. All molecular graphic and video representations were created using PyMOL [66].

## Figures and Tables

**Figure 1 ijms-22-13587-f001:**
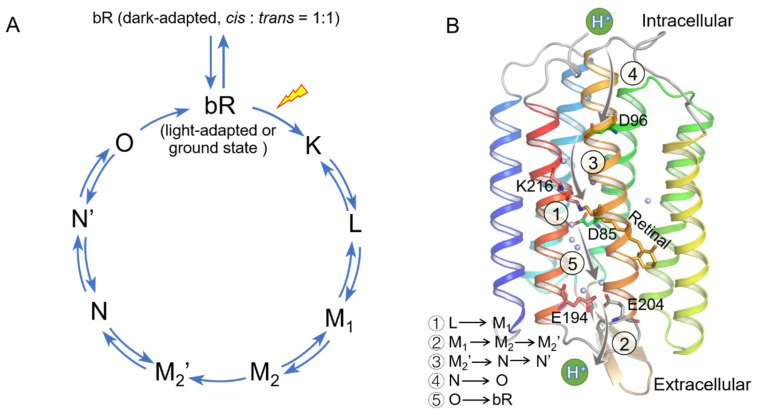
Schematic diagram of bacteriorhodopsin (bR) photocycle and the proton transfer pathway. (**A**) bR photocycle and (**B**) proton transfer pathway. The bR structure of 1C3W (PDB code) is used here. The retinal chromophore possesses a thermal equilibrium between 13-*cis*, 15-*syn* (bR*_cis_*), and all-*trans*, 15-*anti* (bR*_trans_*) isomers in the dark-adapted bR, and light-adapted protein only contains the all-*trans* isomer. Absorption of a photon by bR initiates a catalytic cycle and leads to the vectorial transport of a proton out of the cell through 5 steps. The process can be described as the retinal first photo-isomerizing from all-*trans* to 13-*cis* configuration (light-adapted to K), followed by a proton transfer from the Schiff base (SB) to the proton acceptor D85 (L to M_1_, step 1). Concomitantly, a proton is released to the bulk phase by a group amino acid residue, and the Schiff base changes its accessibility from extracellular to the intracellular direction (M_1_ to M_2_, step 2). The Schiff base is reprotonated from D96 in the cytoplasmic channel to allow vectorial transport of a proton (M_2_ to N, step 3). After reprotonation of D96 from the cytoplasmic surface (N to O, step 4), the retinal isomerizes thermally, and the accessibility of the Schiff base switches back to extracellular (O to light-adapted, step 5). The extracellular part of bR exhibits a hydrogen-bonded network of charged amino acids and water molecules as potential proton translocation pathway elements. Very few charged amino acids and water molecules are located in the cytoplasmic domain [4,5,6,7,8,9,10,11,12,13,14,15,16].

**Figure 2 ijms-22-13587-f002:**
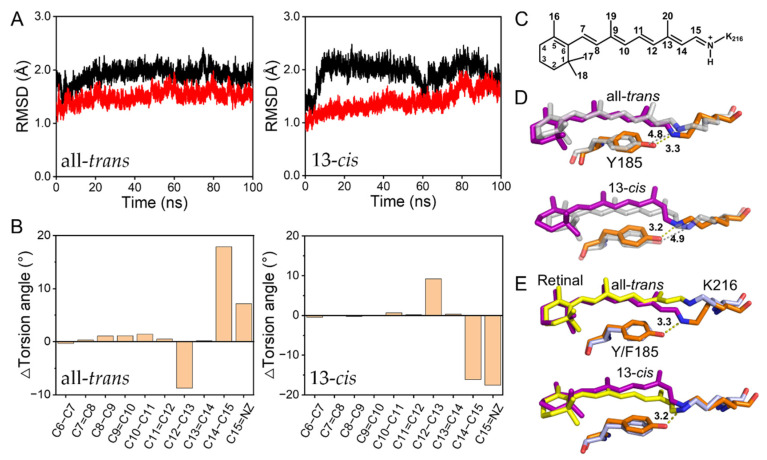
Changes in protein stability, the retinal sidechain torsion angle, and interaction with Y185/F185 in WT-bR and Y185F. (**A**) Time evolutions of root–mean–square deviation (RMSD) of the Cα atoms of the dark-adapted wild-type (black line) and Y185F mutant (red line); the left is bR*_trans_,* and the right is bR*_cis_*. (**B**) The difference in the retinal sidechain torsion angle of Y185F minus WT-bR in the bR*_trans_* and bR*_cis_*, respectively. (**C**) Molecular structure of the retinal Schiff base. (**D**) Crystal (grey, PDB 1C3W for bR*_trans_* and 1X0S for bR*_cis_*) and simulated (purple and orange) structure comparisons of Y185 with the Schiff base in bR*_trans_* and bR*_cis_*. (**E**) The superimposed Y185/F185 with the retinal Schiff base in WT-bR (purple and orange) and Y185F (yellow and light blue) by MD simulations.

**Figure 3 ijms-22-13587-f003:**
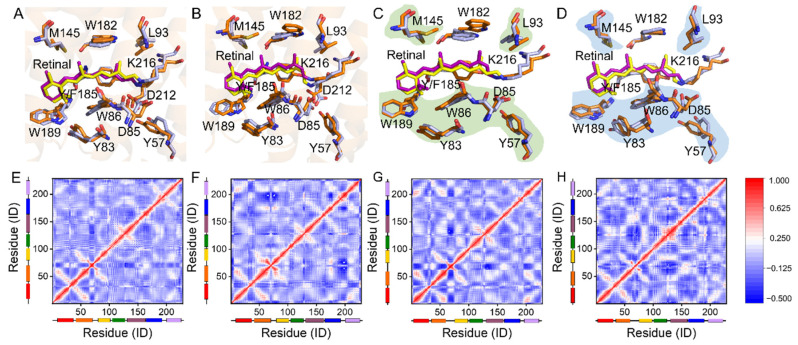
The conformational change in the retinal binding pocket and the interaction of Y185 with retinal in bR*_trans_* and bR*_cis_*. (**A**,**B**) Superimposed retinal binding pocket of WT-bR*_trans_* with Y185F-bR*_trans_*, and WT-bR*_cis_* with Y185F-bR*_cis_*. (**C**,**D**) Superimposed retinal binding pocket of WT-bR*_cis_* with Y185F-bR*_trans_*, and WT-bR*_trans_* with Y185F-bR*_cis_*. Color codings are the same as used in Figure 2E. (**E**–**H**) Dynamical cross-correlated map (DCCM) analyses of the Cα atoms of WT-bR*_trans_*, WT-bR*_cis_*, Y185F-bR*_trans_*, and Y185F-bR*_cis_* by the simulations, respectively.

**Figure 4 ijms-22-13587-f004:**
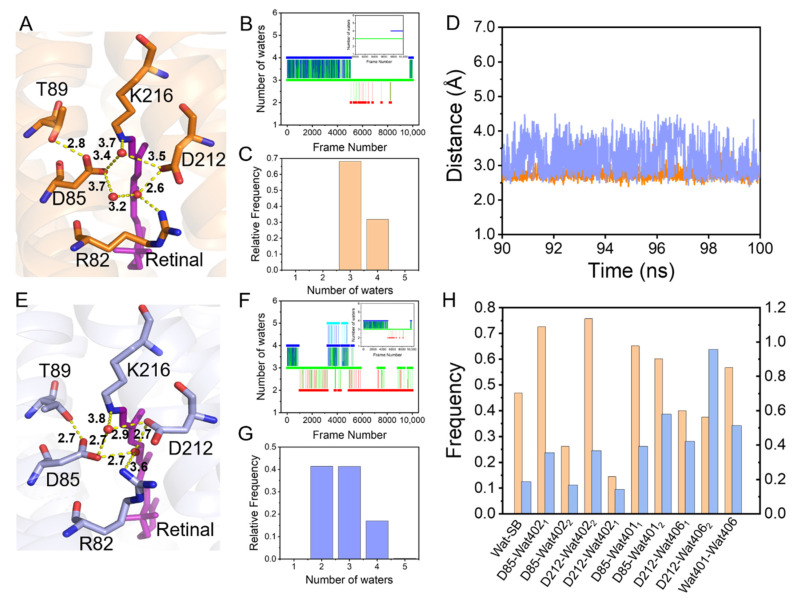
Analyses of the pentagonal hydrogen-bond (H-bond) network in WT-bR and Y185F. (**A**,**E**) Schematic diagrams of the pentagonal hydrogen bonding networks in the ground state of WT-bR and Y185F, respectively. The color coding is the same as in Figure 2E; red spheres represent the water molecules. (**B**,**F**) Fluctuation over time of the number of water molecules around D212, D85, R82, and SB by a distance of 4.0 Å in WT-bR and Y185F-bR; the upper right corner is a partially enlarged view of the last 10 ns MD simulation results. (**C**,**G**) Frequency histogram of water molecules within 4.0 Å around the D212, D85, R82, and SB in WT-bR and Y185F, respectively. (**D**) Time-series of the distances between D85 and T89. (**H**) Histogram of occurrence frequency of the H-bond between D212, D85, and SB with the water molecules in WT-bR and Y185F. Color code for (**D**,**H**): orange and light blue indicate WT-bR and Y185F, respectively.

**Figure 5 ijms-22-13587-f005:**
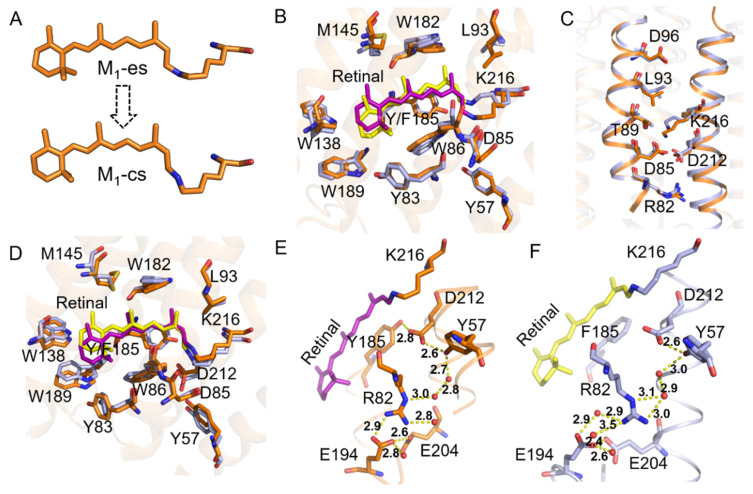
Conformation changes caused by the Y185F mutation in M_1_ and M_2_ states. (**A**) Schematic diagram of the two stages in M_1_ state. (**B**,**C**) Superimposed drawing of the retinal binding pocket residues and the residues within the proton translocation channel in the WT-M_1_-es and Y185F-M_1_-es. (**D**) Superimposed drawing of the key residues in the retinal binding pocket in WT-M_1_-cs and Y185F-M_1_-cs. (**E**,**F**) Schematic diagrams of the H-bond network around the proton release groups in WT-M_2_ and Y185F-M_2_, respectively. The color coding of 4B-F is the same as used in Figure 2E.

**Figure 6 ijms-22-13587-f006:**
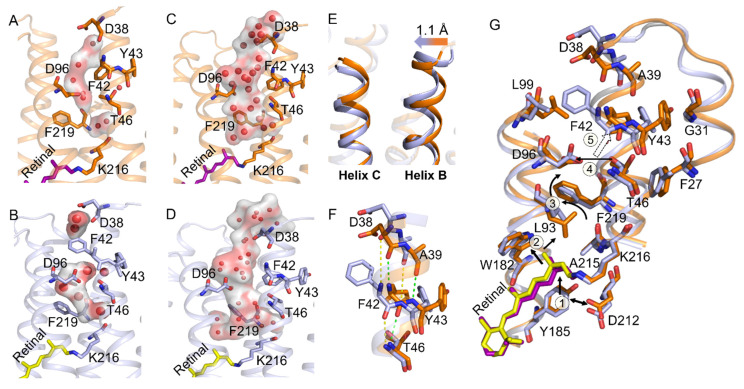
Detailed conformation changes in the cytoplasmic side in M_2_’ and N states caused by the Y185F mutation. (**A**–**D**) Schematic diagrams of the residues and water-chain from the cytoplasmic surface to Schiff base in WT-M_2_’, Y185F-M_2_’, WT-bR_N_, and Y185F-bR_N_, respectively. (**E**) Superimposed conformation of helices B and C in WT-M_2_’ and Y185F-M_2_’. (**F**) Disrupted backbone H-bond pattern around F42 of helix B in Y185F. (**G**) The 5-step sequential conformational changes from SB to F42 by Y185F mutant in M_2_’ state. Color codings are the same as Figure 2E, and the red spheres and transparent surface indicate the water chain.

**Figure 7 ijms-22-13587-f007:**
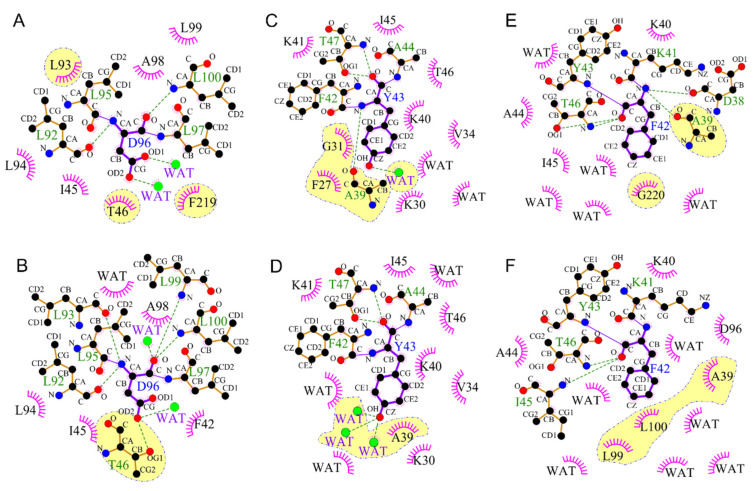
Schematic diagram of the H-bond (green dotted lines) and hydrophobic interactions (pink spoked arcs) around D96 (**A**,**B**), Y43 (**C**,**D**), and F42 (**E**,**F**) in WT-M_2_’ (**A**,**C**,**E**) and Y185F-M_2_’ (**B**,**D**,**F**). Only the van der Waals contacts between the residues with the closest distance less than 4.0 Å are considered.

**Figure 8 ijms-22-13587-f008:**
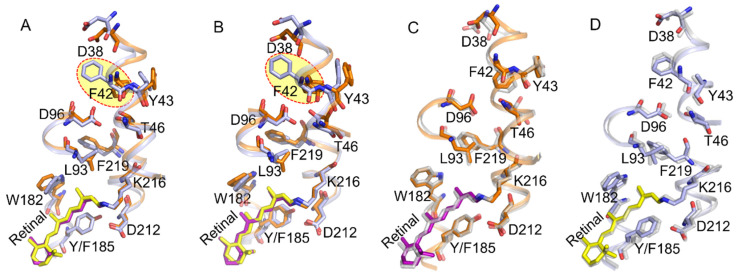
Comparison of the conformation of the cytoplasmic half channel of the M_2_’ state from the repeated simulations. (**A**,**B**) WT-M_2_’ (orange and purple) vs. Y185F-M_2_’ (light blue and yellow) from the second-and the third-time simulations, respectively. Color codings are the same as used in Figure 2E. (**C**,**D**) Superimposed presentations of the second-and the third-time simulations of the cytoplasmic half channel of WT-M_2_’ and Y185F-M_2_’, respectively. The grey color indicates the second-time simulation.

**Figure 9 ijms-22-13587-f009:**
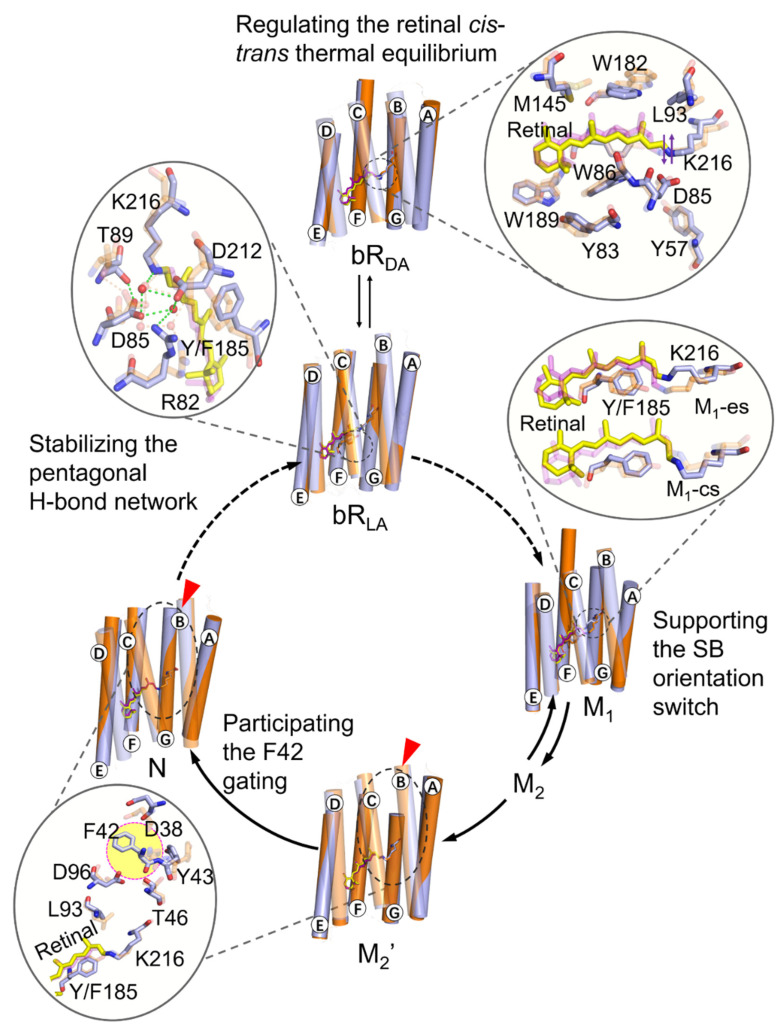
Schematic flowchart of the effects of Y185 on several important events during the bR photocycle. The color coding is the same as in Figure 2E, and the red arrowhead indicates a substantial change in the helix.

**Table 1 ijms-22-13587-t001:** Chemical shifts of the [10, 11, 14, 15-^13^C_4_]-retinal-regenerated WT-bR and Y185F-bR in the dark-adapted and M states as determined by ssNMR and calculated by AF-QM/MM.

	ssNMR [25]				AF-QM/MM				
	C10	C11	C14	C15	C10	C11	C14	C15	RMSE
	WT-bR								
bR*_cis_*	130.0	139.0	110.0	163.2	129.5	138.6	109.5	162.7	0.522
bR*_trans_*	133.2	135.4	122.7	160.0	132.8	135.9	123.2	160.5	0.489
M_1_	129.4	129.4	125.7	164.0	128.9	128.9	125.3	163.5	0.491
M_2_	129.4	129.4	124.0	164.8	128.9	128.9	123.5	164.3	0.504
Y185F-bR									
bR*_cis_*	130.0	138.0	111.8	165.4	129.5	137.6	111.3	165.0	0.428
bR*_trans_*	132.6	135.1	123.0	160.2	132.1	134.7	122.6	159.8	0.438
M_1_	129.5	129.5	125.7	164.0	129.0	129.0	125.2	163.5	0.443
M_2_	129.5	129.5	124.7	165.0	129.0	129.0	124.2	164.5	0.505

**Table 2 ijms-22-13587-t002:** The retinal binding pocket volume of WT-bR*_cis_*, WT-bR*_trans_*, and M_1_ states in the last frame trajectories.

Pocket Volume/Å^3^	Wild-Type	Y185F
bR*_trans_*	947.67	883.40
bR*_cis_*	968.09	989.80
M_1_-es	944.61	985.84
M_1_-cs	977.91	985.53

## Data Availability

Data related to this paper may be requested from the authors.

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
