# Peer review of "Dynamic Coupling of Tyrosine 185 with the Bacteriorhodopsin Photocycle, as Revealed by Chemical Shifts, Assisted AF-QM/MM Calculations and Molecular Dynamic Simulations"

_ijms, 2021, doi:10.3390/ijms222413587_

Round 1
Reviewer 1 Report
This is a very interesting manuscript which suffers from excessive reliance on background knowledge that is not explained in the introduction, and from the. For example: authors mention states M1 and M2 of the bacteriorhodopsin photocycle without first explaining how they differ from each other; the picture of the photocycle they show as last figure is quite illegible and (crucially) does not allow the reader to understand what is happening in the photocycle. There are also several places where the text is quite dense (e.g. in the description of the changes in the binding pocket), which makes information extraction exceedingly difficult. The most important flaw, however, lies in the apparent use of a single simulation of each system. Had the authors found dramatic differences in the immediate vicinity of the regions where the systems differ (retinal C13=C14 bond and residue 185) , the use of a single simulation might be enough to draw conclusions, but that is not the case: differences are quite subtle and I cannot therefore be sure that they are not statistical noise. A detailed breakdown of my impression of the paper follows below:
A) Language is sub-par (e.g. "is to know" (line 16) "can apply" (line 28) etc.etc.)
B) in lines 129-135, authors describe differences between the Y185-K126 distances in simulation vs. crystal structure. A figure comparing the simulated and crystal conformations around those residues would be helpful for the reader here.
C) the data in lines 144-160 are very difficult to follow: they mostly consist of lists of distances which the reader can not possibly relate to each other without referring to the pictures (which in turn makes the text itself redundant). I suggest , instead, that authors show the evolution of those distances along the simulation, since that will allow the reader to immediately see how stable those distances are, how changes in one distance correlate (or not) with changes of other distances, etc., and to provide a table showing the comparison of key distances (+/- std) between WT / mutant and between trans/cis. This same comment applies also to the whole of section 2.3
D) It is not clear to me how the cavity volume was measured: I assume the authors mean "cavity volume AFTER removing the retinal", but that is not made explicit anywhere. In lines 163-165: authors state "The cavity volume of WT-bRcis was smaller than that of WT-bRtrans, consistent with the argument that the decrease of the retinal binding pocket volume can strengthen the 13-cis isomer" . Here I believe authors mean "strengthen the ability to bind the 13-cis isomer", but regardless I cannot see how the decrease in pocket volume can increase binding affinity to one of the isomers, how the volume of the cavity itself gives any information regarding the stabilization of each isomer form, or how the authors establish the direction of any causality relationships. Moreover, since the simulations were performed WITH retinal I cannot quite see how authors can disentangle the effect of the protein-retinal interactions from the effect of the Y185F mutation itself on the empty cavity volume.
E) In figure 2, the legend to panels B and D is the same ("[Superposition..] of WT-bRtrans with Y185F-bRcis").
F) Authors claim that their DCCM plots show differences in dynamical behavior between WT and mutant. The plots are, however, virtually indistinguishable, and I cannot see how the text in lines 175-177 can therefore be supported ("Removal of the Y185–O·····SB–NH H-175 bond changed the dynamic feature of the whole protein and eventually weakened the proton translocation function of the protein") . The DCCM plots also seems to be wrong: it states "Dynamical cross-correlated map (DCCM) analyses of the Cα atoms during simulations for bRtrans and bRcis, respectively, the lower right triangle is wild type (WT), and the upper left triangle is Y185F mutant" , which would mean 2 panels would be enough to show the whole data, but 4 panels are present instead. I also do not think that comparisons of these plots can give us any actionable information: since only one simulation was performed for each state, there is no way to know whether any subtle differences between plots are due to the differences in retinal conformation and WT/mutant differences or are simply random fluctuations similar to what one would observe when running replicate simulations from the same starting condition. This comment applies to all other comparisons that authors detail (e.g. the pentagonal H-bond network, opening the F42 gate in the M2' to N states, etc.)
G) Authors suggest that Y185 influences the orientation of F42 in the M2' to N states. Those residues are quite far apart (18 angstrom), and such a suggestion therefore requires much more support than the comparison of a single simulation of WT with a single simulation of Y185F: how can we tell whether the difference is not a simple statistical happenstance otherwise?
Author Response
Reviewer #1:
Comments and Suggestions for Authors: This is a very interesting manuscript which suffers from excessive reliance on background knowledge that is not explained in the introduction, and from the. For example: authors mention states M1 and M2 of the bacteriorhodopsin photocycle without first explaining how they differ from each other; the picture of the photocycle they show as last figure is quite illegible and (crucially) does not allow the reader to understand what is happening in the photocycle. There are also several places where the text is quite dense (e.g. in the description of the changes in the binding pocket), which makes information extraction exceedingly difficult. The most important flaw, however, lies in the apparent use of a single simulation of each system. Had the authors found dramatic differences in the immediate vicinity of the regions where the systems differ (retinal C13=C14 bond and residue 185) , the use of a single simulation might be enough to draw conclusions, but that is not the case: differences are quite subtle and I cannot therefore be sure that they are not statistical noise.
Answer: We gratefully appreciate your critical comments and valuable suggestions, which are helpful for us to improve the quality of the manuscript. We have expanded the background information of the bR photocycle and proton transfer passway, and have added a figure (Figure S1) in the Supplementary Materials (SM). We have explained the M1 (early M) and M2 (late M) states in section 2.3. Our results showed that Y185 plays a significant role in regulating the retinal cis-trans thermal equilibrium, stabilizing the pentagonal H-bond network, participating in the orientation switch of Schiff Base nitrogen, and opening of the F42 gate by interacting with the retinal and several key residues along the proton translocation channel. The last figure (Figure 7) is a schematic diagram describing the functional roles of Y185 during the bR photocycle based our simulation results. We have simplified the dense text in lines 144-160 and section 2.3, and have added more diagrams and tables to support our arguments (Figures S7, S10-S13, Table S5-S9). Please see the main text and SM for the corresponding changes. We ran simulations at least twice for each examined system to make sure that the changes observed in this work are not statistical accidents. We have added some outcomes from the repeated (second or third) simulations in the SM for comparison (Figure S15-S18, Table S5-S9).
- A) Language is sub-par (e.g. "is to know" (line 16) "can apply" (line 28) etc.etc.)
Answer: Thank you for your criticism. We have corrected the spelling and grammar errors and used the editing services to polish further the language (https://www.mdpi. com /authors/english).
- B) in lines 129-135, authors describe differences between the Y185-K126 distances in simulation vs. crystal structure. A figure comparing the simulated and crystal conformations around those residues would be helpful for the reader here
Answer: Thank you for your kind suggestion. We have added a figure below (Figure S6) to compare the simulated and crystal conformations around Y185-K216 in SM.
- C) the data in lines 144-160 are very difficult to follow: they mostly consist of lists of distances which the reader can not possibly relate to each other without referring to the pictures (which in turn makes the text itself redundant). I suggest , instead, that authors show the evolution of those distances along the simulation, since that will allow the reader to immediately see how stable those distances are, how changes in one distance correlate (or not) with changes of other distances, etc., and to provide a table showing the comparison of key distances (+/- std) between WT/mutant and between trans/cis. This same comment applies also to the whole of section 2.3
Answer: Thank you for your great suggestion. We have simplified the lines 144-160 and section 2.3 dense text descriptions about the change of distances, and have added diagrams and tables to show the evolution of those distances along with the simulation time in SM (Figures S7-S8 and S11-S13, and Table S5-S9).
- D) It is not clear to me how the cavity volume was measured: I assume the authors mean "cavity volume AFTER removing the retinal", but that is not made explicit anywhere. In lines 163-165: authors state "The cavity volume of WT-bRcis was smaller than that of WT-bRtrans, consistent with the argument that the decrease of the retinal binding pocket volume can strengthen the 13-cis isomer". Here I believe authors mean "strengthen the ability to bind the 13-cis isomer", but regardless I cannot see how the decrease in pocket volume can increase binding affinity to one of the isomers, how the volume of the cavity itself gives any information regarding the stabilization of each isomer form, or how the authors establish the direction of any causality relationships. Moreover, since the simulations were performed WITH retinal I cannot quite see how authors can disentangle the effect of the protein-retinal interactions from the effect of the Y185F mutation itself on the empty cavity volume.
Answer: We are very sorry for the misleading. The word “cavity” we used here refers to the volume size of the retinal binding pocket, it composes of retinal and the key residues around it suitable for binding in our calculation (below). In the dark-adapted bR, the retinal chromophore possesses a cis-trans thermal equilibrium, and it was reported that the retinal binding pocket volume size reflects a preference for the retinal configuration (Tsuda et al. Biophys. J. 30:149-157 (1980); Schulte et al. Biophys. J. 69:1554-1562 (1995); Bryl et al. Eur. Biophys. J. 31:539-548 (2002)).
Many software and methods are suitable for calculating the volume of a ligand-binding pocket in the protein. We used the pocket detection plugin (Pck 2.0.3) of VMD in this paper (Humphrey et al. J. Mol. Graphics. 14:33-38 (1996); Edelsbrunner et al. Discrete Appl. Math. 1998, 88:83-102 (1998); Edelsbrunner et al. Proc. Natl. Acad. Sci. U.S.A. 100:2203-2208 (2003)). Please refer to the website (http://schwarz.benjamin. free.fr/Work/Pck/manual.htm) for more information. We never removed the retinal from its binding pocket for the calculation, so the difference in the pocket size of WT-bR and Y185F should be caused by the Y185F mutation.
- E) In figure 2, the legend to panels B and D is the same ("[Superposition..] of WT-bRtrans with Y185F-bRcis").
Answer: Thanks for pointing it out, and we have corrected it.
- F) Authors claim that their DCCM plots show differences in dynamical behavior between WT and mutant. The plots are, however, virtually indistinguishable, and I cannot see how the text in lines 175-177 can therefore be supported ("Removal of the Y185–O·····SB–NH H-175 bond changed the dynamic feature of the whole protein and eventually weakened the proton translocation function of the protein"). The DCCM plots also seems to be wrong: it states "Dynamical cross-correlated map (DCCM) analyses of the Cα atoms during simulations for bRtrans and bRcis, respectively, the lower right triangle is wild type (WT), and the upper left triangle is Y185F mutant" , which would mean 2 panels would be enough to show the whole data, but 4 panels are present instead. I also do not think that comparisons of these plots can give us any actionable information: since only one simulation was performed for each state, there is no way to know whether any subtle differences between plots are due to the differences in retinal conformation and WT/mutant differences or are simply random fluctuations similar to what one would observe when running replicate simulations from the same starting condition. This comment applies to all other comparisons that authors detail (e.g. the pentagonal H-bond network, opening the F42 gate in the M2' to N states, etc.)
Answer: We appreciate your critical comments and address the questions as below.
1) We changed the DCCM plots but forgot to update the legend. We feel very sorry for our carelessness, and the mistake has been corrected in the revised manuscript.
2) We agree that the Y185F mutation should not cause a dramatic change to the whole protein dynamics. However, conformation dynamics are fundamental for bR to transfer a proton across the cell membrane through rapid response between distant structural domains by various interactions. We think Y185F mutation might disrupt bR dynamics from a more coherent mode to a more incoherent mode, and DCCM can provide a good indication, although the DCCM plots we showed here do not present a dramatic change. Without disrupting the overall protein coherent dynamics, we should not have seen disrupted M, N, and O states in Y185F (Ding et al. Biochim. Biophys. Acta 1857:1786-1795 (2016); Ding et al. Biochim. Biophys. Acta 1859:1006-1014 (2018)).
3) We ran simulations at least twice for each examined system to make sure that the changes observed in this work are not statistical accidents.
- G) Authors suggest that Y185 influences the orientation of F42 in the M2' to N states. Those residues are quite far apart (18 angstrom), and such a suggestion therefore requires much more support than the comparison of a single simulation of WT with a single simulation of Y185F: how can we tell whether the difference is not a simple statistical happenstance otherwise?
Answer: We appreciate your critical comments. Although Y185 and F42 are far from each other, such perturbation from the extracellular side to the cytoplasmic side is unfavorable. However, many studies have demonstrated the existence of such long-range effects. For example, Bondar et al. proposed that perturbations caused by mutations in remote regions of the SecY translocons could be relayed to the plug rapidly, causing its displacement and increasing hydration (Structure 18:847-857 (2010)). Tanio et al. demonstrated the D85-V49 long-distance effect has an impact on the bR backbone conformation (Biophys. J. 77:431-442 (1999)). Luecke et al. proposed that M decay occurs when several residues and water molecules in the cytoplasmic side form a transient proton-conducting network from the surface to SB (Science 286:255-260 (1999)). del Val et al. reported that the retinal binding residue D212 has long-range effects on water and hydrogen-bonding dynamics of the cytoplasmic side, and the D96-T46T47(DTT) motif is more sensitive to long-range perturbations (J. Struct. Biol. 186:95-111 (2014)). So, the long-range mediation of Y185 to F42 is quite unusual but still possible.
To rule out the statistical happenstance from a single simulation, we repeated the simulation 3 times to confirm that the observations we had are not statistical noise. The second- and third-time simulation results also showed that the F42 gate opened in WT-Mn and closed in Y185F-Mn (Figure S18). We also led the repeated simulation results of the distance between Y185 and Schiff base (Figure S15), torsion angle of the retinal side sidechain (Figure S16, Table S2), the pentagonal hydrogen-bond network (Figure S17), and the orientation of F42 (Figure S18) in SM for further comparison.

Reviewer 2 Report
The authors investigate the coupling of Tyrosine 185 dynamics with the BR photocycle using a combined method of quantum mechanics/molecular mechanics calculations and molecular dynamics simulations, which are based on chemical shifts obtained by solid-state NMR. The results show that tyrosine 185 plays an important role throughout several intermediates of the photocycle, e.g. by regulating the retinal cis-trans thermal equilibrium or by opening of the F42 gate. The study is original and generally well-written, the method combining theory with experiment proves to be highly fruitful. As such, the paper can almost be accepted as is.
I only wonder why a dynamics study of the BR photocycle in a special issue about structural dynamics does not mention important work using quasielastic neutron scattering for investigations of BR protein dynamics.
Author Response
Reviewer #2:
Comments and Suggestions for Authors: The authors investigate the coupling of Tyrosine 185 dynamics with the BR photocycle using a combined method of quantum mechanics/molecular mechanics calculations and molecular dynamics simulations, which are based on chemical shifts obtained by solid-state NMR. The results show that tyrosine 185 plays an important role throughout several intermediates of the photocycle, e.g. by regulating the retinal cis-trans thermal equilibrium or by opening of the F42 gate. The study is original and generally well-written, the method combining theory with experiment proves to be highly fruitful. As such, the paper can almost be accepted as is.
I only wonder why a dynamics study of the BR photocycle in a special issue about structural dynamics does not mention important work using quasielastic neutron scattering for investigations of BR protein dynamics.
Answer: We are grateful for your comments, and have cited and discussed the following related work in the revised manuscript.
- Ferrand, M., Dianoux, A.J., Petry, W. & Zaccaï, G. Thermal motions and function of bacteriorhodopsin in purple membranes: effects of temperature and hydration studied by neutron scattering. Proceedings of the National Academy of Sciences of the United States of America 90, 9668-9672 (1993).
- Ferrand, M. et al. Structure and dynamics of bacteriorhodopsin. Comparison of simulation and experiment. FEBS Letters 327, 256-60 (1993).
- Fitter, J., Lechner, R.E., Buldt, G. & Dencher, N.A. Internal molecular motions of bacteriorhodopsin: hydration-induced flexibility studied by quasielastic incoherent neutron scattering using oriented purple membranes. Proceedings of the National Academy of Sciences of the United States of America 93, 7600-7605 (1996).
- Fitter, J., Lechner, R.E. & Dencher, N.A. Picosecond molecular motions in bacteriorhodopsin from neutron scattering. Biophysical Journal 73, 2126-2137 (1997).
- Réat, V. et al. Dynamics of different functional parts of bacteriorhodopsin: H-2H labeling and neutron scattering. Proceedings of the National Academy of Sciences of the United States of America 95, 4970 (1998).

Round 2
Reviewer 1 Report
I think the presentation is still far from optimal, becuase authors have relegated key pieces of information to the SM: Current figures S1 and S6 should be moved to the main text. I also think that a summary of the results from the three simulations should be presented in the main text. In this regard, I find the superposed pictures provided by the authors (especially fig S18 to be hard to interpret (which is compounded by the unclear wording of the Figure captions). Regarding the F42 flip, I think that the simulations may be able to shed light on the mechanism underlying it: a graph showing the evolution of specific distances (throughoutbthe whole simultion) in the cytoplasmic half-channel between retinal-F42 (e.g. F/Y185 to SB, SB to C=O, C=O to the following peptide bond NH, or SB to water, etc.) might give clues regarding the order of the events leading to the change of conformation , and tell us whether the flip is due to (for example) an interaction(or lack thereof) of F/Y185 with the helix, which is then propagated, or is instead due to a change in the flexibility of the SB, wich enable SB to "push" (or "pull") the helix upwards or downards, etc.
Author Response
Response to Reviewer #1:
Comments and Suggestions for Authors: I think the presentation is still far from optimal, because authors have relegated key pieces of information to the SM: Current figures S1 and S6 should be moved to the main text. I also think that a summary of the results from the three simulations should be presented in the main text. In this regard, I find the superposed pictures provided by the authors (especially fig S18 to be hard to interpret (which is compounded by the unclear wording of the Figure captions). Regarding the F42 flip, I think that the simulations may be able to shed light on the mechanism underlying it: a graph showing the evolution of specific distances (throughout the whole simulation) in the cytoplasmic half-channel between retinal-F42(e.g. F/Y185 to SB, SB to C=O, C=O to the following peptide bond NH, or SB to water, etc.) might give clues regarding the order of the events leading to the change of conformation, and tell us whether the flip is due to (for example) an interaction(or lack thereof) of F/Y185 with the helix, which is then propagated, or is instead due to a change in the flexibility of the SB, which enable SB to "push" (or "pull") the helix upwards or downwards, etc.
Answer: We very much appreciate your constructive comments and suggestions and have made the following changes accordingly:
1) We have moved Figures S1 and S6 to the main text.
2) The purpose of repeating the simulations used in the main text is to ensure that the results we showed in the main text are not statistical accidents, and most of the figures in the main text are from the first-time simulation. We have put more results of the second-and third-time simulations in the main text and SM for comparison.
3) We have put the repeated simulation results of the changes in distance between Y185 and Schiff base, torsion angle of the retinal sidechain, and the pentagonal hydrogen-bond network in SM.
4) In the previous manuscript, Figure S18 showed an overlay of the second-and third-time simulation results of WT and Y185F mutant in the M2’ state, confirming the long-range regulation of Y185 to F42 is not accidental. To further enhance the integrity of the main text, we have moved Figure S18 from SM to Figure 8 of the main text and modified the wording of the Figure caption accordingly.
5) We have added, and modified diagrams and tables showing the time evolution of specific distances in the cytoplasmic half-channel from the retinal to F42 (Figure S16, Table S10) and presented the sequential conformation changes in the cytoplasmic half channel from SB to F42 (Figure 6). Please refer to the main text for a detailed description of the events.
6) We have explained more about the repeated simulations in the corresponding parts of the main text (lines 160-164 and 243-244).

Round 3
Reviewer 1 Report
I still think the presentation is far from optimal: specifically, the graphs showing the evolution of distances in Figure S16 should depict the evolution of the distances throughout the whole simulation ( to accurately show wihich distances change first) not only the last few ns.
